# The Curcumin Supplementation with Piperine Can Influence the Acute Elevation of Exercise-Induced Cytokines: Double-Blind Crossover Study

**DOI:** 10.3390/biology11040573

**Published:** 2022-04-10

**Authors:** Stéfani Miranda-Castro, Felipe J. Aidar, Samara Silva de Moura, Lucas Marcucci-Barbosa, Lázaro Fernandes Lobo, Francisco de Assis Dias Martins-Júnior, Roberta da Silva Filha, Pedro Alves Soares Vaz de Castro, Ana Cristina Simões e Silva, Danielle da Glória de Souza, Siomara Aparecida da Silva, Kelerson Mauro de Castro Pinto, Guilherme de Paula Costa, Ana Filipa Silva, Filipe Manuel Clemente, William Valadares Campos Pereira, Albená Nunes-Silva

**Affiliations:** 1Laboratory of Exercise Inflammation and Immunology, School of Physical Education, Federal University of Ouro Preto (LABIIEX/EEF-UFOP), Ouro Preto 35400-000, Brazil; stefani_ef@yahoo.com.br (S.M.-C.); lucasmarcucci@gmail.com (L.M.-B.); lflobo7@gmail.com (L.F.L.); francisco.junior@aluno.ufop.edu.br (F.d.A.D.M.-J.); kelerson2@yahoo.com.br (K.M.d.C.P.); valadareswcp@gmail.com (W.V.C.P.); 2Graduate Health and Nutrition Program, Federal University of Ouro Preto (PPGSN/UFOP), Ouro Preto 35400-000, Brazil; samara_silva09@hotmail.com; 3Department of Physical Education, Federal University of Sergipe (UFS), São Cristóvão 49100-000, Brazil; fjaidar@gmail.com; 4Interdisciplinary Laboratory of Medical Investigation, Department of Pediatrics, Faculty of Medicine UFMG (LIIM/UFMG), Belo Horizonte 30130-100, Brazil; rosilva_beta@hotmail.com (R.d.S.F.); pedroasvc@gmail.com (P.A.S.V.d.C.); acssilva@hotmail.com (A.C.S.e.S.); 5Department of Microbiology, Institute of Biological Sciences, Federal University of Minas Gerais (UFMG), Belo Horizonte 31270-901, Brazil; souzadg@gmail.com; 6Sports Teaching Methodology Laboratory, School of Physical Education, Federal University of Ouro Preto (LAMEES/EEF-UFOP), Ouro Preto 35400-000, Brazil; siomarasilva@ufop.edu.br; 7Graduate Program in Biological Sciences, Research Center in Biological Sciences, Federal University of Ouro Preto (CEBIOL), Ouro Preto 35400-000, Brazil; gpcostabio@gmail.com; 8Sports and Leisure School, Polytechnic Institute of Viana do Castelo, Rua Escola Industrial e Comercial de Nun’Álvares, 4900-347 Viana do Castelo, Portugal; anafilsilva@gmail.com (A.F.S.); filipe.clemente5@gmail.com (F.M.C.); 9Research Center in Sports Performance, Recreation, Innovation and Technology (SPRINT), 4960-320 Melgaço, Portugal; 10The Research Centre in Sports Sciences, Health Sciences and Human Development (CIDESD), 5001-801 Vila Real, Portugal; 11Instituto de Telecomunicações, Delegação da Covilhã, 1049-001 Lisboa, Portugal

**Keywords:** curcumin, inflammation, running, dietary supplements

## Abstract

**Simple Summary:**

Inflammation is an immune response to harmful stimuli, such as pathogens and damaged cells. Intense exercise can induce a local and systemic inflammatory response as well. We believe that exercise-induced inflammatory responses are essential for muscle repair and regeneration. However, uncontrolled acute inflammation in athletes during the training process and competition could reduce the level of performance. In this study, we investigate the effects of curcumin and piperine on the exercise-induced inflammatory response.

**Abstract:**

Background: to evaluate the effects of one week of supplementation with curcumin combined with piperine on physical performance, immune system cell counts, muscle damage, and plasma levels of inflammatory markers after a treadmill running training session. Methods: This study is a double-blind, crossover-balanced clinical trial with a three-week intervention. Sixteen male runners with a mean age of 36 ± 9 years and VO2 max of 60.6 ± 9.03 mL.kg ^−1^ min ^−1^ were recruited and randomly divided into 2 groups: the first group (CPG) was supplemented daily for 7 days with 500 mg of curcumin + 20 mg piperine, and the second group (PG) was supplemented with 540 mg of cellulose. After the 7th day of supplementation, the volunteers participated in the experimental running protocol, where blood samples were collected before, after, and one hour after exercise for analysis of the number of leukocytes, creatine kinase, and cytokine concentration (IL-2, TNF-α, IFN, IL-6, and IL-10) using flow cytometry. This process was repeated, reversing the supplementation offered to the groups. Results: curcumin and piperine supplementation could not change the physical performance, immune cell counts, and muscle damage; however, the aerobic fatiguing exercise protocol inhibited the elevation of the plasmatic levels of some cytokines. The running exercise protocol could elevate the circulating levels of IL-2 (from 49.7 to 59.3 pg/mL), TNF-α (from 48.5 to 51.5 pg/mL), INF (from 128.8 to 165.0 pg/mL), IL-6 (from 63.1 to 77.3 pg/mL), and IL-10 (from 48.9 to 59.6 pg/mL) 1 h after the end of the running protocol. However, the curcumin and piperine supplementation could inhibit this elevation. Conclusions: curcumin and piperine supplementation had no effect on physical performance, immune cell counts, or muscle damage; however, the supplementation could modulate the kinetics of IL-2, TNF-α, INF, IL-6, and IL-10 1 h after the end of exercise.

## 1. Introduction

The incessant search for techniques and methods to maximize physical performance is extensive in elite sports. Constantly, athletic coaches and the scientific community seek supplements with beneficial ergogenic effects for increased performance in exercises [1]. In this way, several bioactive compounds present in foods that have antioxidant and anti-inflammatory actions have been commonly used as a strategy to accelerate the regeneration process, delay fatigue, and even enhance physical performance [2,3,4]. In this regard, the association between curcumin and piperine may present an ergogenic aid in sports performance, based on their antioxidant, antifungal, and anti-inflammatory functions [5,6]. In addition, this association may contribute to the diversity of the intestinal microbiota, delaying aging, and increasing human life expectancy [7].

During running, the mechanical stress results in muscle damage, triggering a local inflammatory response that activates the nuclear transcription factor-κB (NF-κB) [8]. These effects may worse physical performance if the inflammation is prolonged [9]. Piperine can increase the uptake of curcumin [10] and both reduce NF-κB activation by binding the activator protein-1 (AP-1) to DNA and may reduce the production of the enzyme cyclooxygenase-2 (COX-2), which plays a key role in inflammation [11,12,13,14]. Another pathway up-regulated by curcumin is the Janus-kinase signal transducer and activator of protein transcription (JAK/STAT), resulting in reduced production of interleukin-1 (IL-1), interleukin-2 (IL-2), interleukin-6 (IL-6), interleukin-8 (IL-8), interleukin-12 (IL-12), tumor necrosis factor-alpha (TNF-α), and monocyte chemoattractant protein-1 (MCP-1) [5,13].

Inflammation is a biological response of the immune system that prevents, limits, and repairs damage by invading pathogens or endogenous biomolecules. Although acute inflammation is a transient inflammatory response and is beneficial to the organism, including during recovery after sports sessions or competition, a persistent inflammatory response is associated with tissue dysfunction and pathology. In this sense, strategies that help in the recovery period have been studied.

Despite studies on the benefits of curcumin in several areas of health [5,11,12,14,15], little is known about the effects on runners. Our hypothesis is that, considering the relationship between physical exercise and the immune system, supplementation with curcumin associated with piperine may result in muscle regeneration and improvement in athletic performance in runners. Therefore, the objective of this study was to evaluate the effect of supplementation of curcumin associated with piperine in runners after a treadmill running session until voluntary fatigue for the following parameters: physical performance, immune system cells, muscle damage, and circulating levels of inflammatory mediators

## 2. Materials and Methods

### 2.1. Study Design and Ethical Approval

Sixteen trained male runners volunteered to participate in a double-blind randomized study after signing the terms of consent. Participants were: 36 ± 9 years, body mass of 69.65 ± 7.63 kg, height of 175.9 ± 0.05 cm, peak oxygen consumption of 60.6 ± 9.03 mL/kg/min, and fat percentage of 12.78 ± 3.95. The Ethics Committee of the Federal University of Ouro Preto approved the study under number 4144351, CAAE 31073520.1.0000.5150. The study was developed in accordance with the resolution 466/2012 of the National Research Ethics Commission–CONEP, of the National Health Council, in accordance with the ethical principles expressed in the Helsinki Declaration (1964, reformulated in 2013), by the World Medical Association.

### 2.2. Study Protocol

The participants came to the laboratory four times (M1; M2; M3; M4) (Figure 1—experimental design) as follows: (M1)—the participants performed the anthropometric evaluation, peak VO2 test, and application of the first supplementation. Participants answered a physical fitness capacity form that includes weekly running frequency and volume, history of injury and illness, and medication use in the 4 weeks prior to the start of the study. (M2)—they performed the first experimental trial followed by a 7-day washout. In (M3)—the participants performed the cross-over, starting the second week of supplementation, and then (M4) when the participants performed the second experimental trial. The anthropometric evaluation was determined by means of a digital scale (FILIZOLA^®^, São Paulo, Brazil) with an accuracy of 0.1 kg. For height measurement, a stadiometer with a scale of 0.1 cm (WISO^®^) was used. The fat percentage was evaluated by means of the skinfold technique and was calculated according to the protocol used by Jackson and Pollock (1978) [16]. Next, the volunteers completed a peak VO2 test, using open-circuit spirometry in VO2000^®^ equipment (VO2000, Med Graphics^®^, Saint Paul, MN, USA) through the maximum incremental protocol on a treadmill [17]. The protocol consisted of a warm-up at 5.0 km/h with 5% inclination for 5.0 min, and from this time, the speed was increased by 1.0 km/h every minute until voluntary fatigue.

After the test, subjects received—in a randomized, double-blind fashion—a pot containing 14 capsules that may or may not include 250 mg of curcumin extract associated with 10 mg of piperine extract (CPG), or a placebo (PG) with 260 mg of cellulose. Subjects were instructed to consume 1 capsule at lunch and 1 capsule at dinner for seven consecutive days. The capsules were identical in size, appearance, and smell, composed of vegetable gelatin stained by chlorophyll, making it impossible for the subjects to distinguish the contents.

At the second meeting, after the previous ingestion of supplementation (minimum of 12 h before the collection), the subjects were submitted to the incremental running protocol on a treadmill, which used the peak VO2 collected at M1 as a parameter. The test started with a warm-up phase, with an increased intensity of 40% of VO2 peak for five minutes, which then was increased to 80% of VO2 peak until voluntary fatigue. Next, for the recovery phase, the intensity was reduced to 40% of VO2 peak for 5 min. During the running protocol, the subjects did not receive feedback related to elapsed time, distance traveled, or heart rate, and no fluids were ingested. In the 24 h prior to the first experimental trial, subjects recorded all food and fluid ingestions, replicating these patterns the day before the second test. Venous blood was collected before, immediately after and 1 h after the exercise. The washout period lasted seven days.

In the third meeting, which occurred seven days after M2, the crossover was performed, initiating the second week of supplementation (contrary to M2). In M4, the volunteers once again participated in all the procedures adopted in M2. During the intervention protocol, it was requested that physical training, food intake, and sleep hours were kept consistent. The individuals were also instructed not to use food supplements, compounds that may contain curcumin or piperine, or anti-inflammatory medications. In addition, they were recommended to abstain from strenuous exercise in the 48 h preceding the battery of physical tests. Lastly, the experimental trials of all participants were conducted in the morning, maintaining the same hours and conditions during the two days of tests [18].

### 2.3. Blood Sampling and Laboratory Measurements

A trained and qualified phlebotomist performed venipuncture with appropriate materials and respected all biosafety procedures. Blood samples (4 mL) were collected from the cubital vein and placed in two tubes containing EDTA. Then, one tube was used for the blood count, and the other tube was centrifuged at a speed of 3000 rpm for 10 min. Aliquots of plasma were transferred to Eppendorf tubes and stored in a −80 C° freezer until analysis.

The concentrations of cytokines IL-2, TNF-α, interferon (IFN), IL-6, and IL-10 were analyzed by flow cytometry technique using a FACS Canto II flow cytometer (BD Biosciences, San Jose, CA, USA). The Human Th1 Cytokine Kit Panel (5-plex) w/PFV02-Legendplex TM was used, following the manufacturer’s recommendations. CK analysis was performed using the Creatine Kinase Kit (Reference 07190794 190) using a Cobas Integra 400 plus equipment (Roche, Sandhofer, Mannheim, Germany), according to the manufacturer’s recommendations. A complete blood count was performed using the bench-top hematology analyzer model Sapphire from CELL-DYN, as recommended by the manufacturer.

### 2.4. Statistical Analysis

The statistical analyses were conducted using the GraphPad Prism 8.4.2 statistical program. The normality of the data was verified by the Shapiro–Wilk test. To evaluate the running performance between the two groups, the Wilcoxon test was used. To compare the concentrations of both cytokines and CK between both the groups and the times, two-way ANOVA and Tukey’s post hoc tests were both used. Finally, to compare the number of cells of the immune system between the groups and between the times, the Friedman test was applied. The level of significance adopted was *p* ≤ 0.05. To analyze the effect size, Cohen’s test was used.

## 3. Results

To evaluate the physical performance (Figure 2), the distance reached (Figure 2a) and the running duration time (Figure 2b) were compared between the curcumin associated with the piperine group (CPG) and the placebo group (PG). There were no significant differences between the groups in both variables.

The effect of curcumin associated with piperine extract supplementation on immune system cell counts was evaluated at different times (pre, post, and 1 h after the experimental running protocol). An increase in the number of leukocytes immediately after running was observed in a similar way in the CPG (*p* = 0.003) and PG (*p* = 0.007) groups.

No statistical differences were found between the groups at 1 h after exercise (Figure 3a). The number of circulating neutrophils also increased immediately after exercise in CPG (*p* < 0.001) and PG (*p* < 0.001). The same is true for 1 h after the experimental protocol in both groups: CPG (*p* < 0.001) and PG (*p* < 0.001) (Figure 3b). The number of monocytes did not change after the running protocols in both groups (Figure 3c). Lymphocyte counts immediately after exercise increased similarly between the CPG (*p* < 0.001) and PG (*p* < 0.001) groups, followed by a decrease at 1 h after the running protocol compared to pre-exercise levels: CPG (*p* < 0.001) and PG (*p* < 0.001) (Figure 3d).

Figure 4 shows the creatine kinase (CK) levels. The running protocol altered the CK concentration, in a similar way, in the CPG (*p* < 0.001) and PG (*p* = 0.003) groups at the post-exercise time (Figure 4). However, there was no significant difference between the groups.

In this study (Figure 5), the running protocol elevated the concentrations of all cytokines in the PG group. There were significant increases when comparing the time before versus 1 h after exercise in the concentrations of IL-2 (Figure 5a), TNF-α (Figure 5b), IFN (Figure 5c), IL-6 (Figure 5d), and IL-10 (Figure 5e). Regarding the curcumin associated with the piperine-supplemented group, the concentrations of IL-2 and TNF-α significantly reduced 1 h after exercise if compared to the levels before (*p* = 0.04 and *p* = 0.03, respectively). No significant changes were detected for IFN, IL-6, and IL-10 concentrations in CPG when comparing time points. However, there were significant differences in cytokine concentrations between the groups at 1 h after the exercise. Levels of IL2 (*p* = 0.013), TNF-α (*p* = 0.004), IFN (*p* = 0.034), and IL-6 (*p* = 0.038) were significantly lower in CPG than in PG.

## 4. Discussion

This study aimed to evaluate the effect of curcumin supplementation associated with piperine on immune system cells, physical performance, muscle damage, and blood inflammatory markers after a running session until voluntary fatigue.

The most important findings of this study were the administration of curcumin associated with piperine can inhibit the elevation of plasmatic levels of some cytokines such as IL-2, TNF-α, IFN, IL-6, and IL-10. This result shows the capacity of curcumin to modulate an inflammatory response acutely induced by exercise.

The search for alternatives that can improve physical performance is the focus of several scientific studies [19,20,21]. Some authors [4] suggest that associated curcumin supplementation can improve physical performance in humans. In this project, this hypothesis was avoided. Probably in the training process scenario, where the time and quality of recovery are important to keep a high level of performance during some period, the curcumin supplementation could be relevant.

However, in the present study, the supplementation of 500 mg of curcumin associated with 20 mg of piperine consumed for 7 days was not able to increase physical performance in the exercise protocol, as evaluated by the distance walked and running time.

Regarding total leukocyte count, we observed a significant and similar increase when values immediately after exercise were compared to those before in both groups. We also detected a statistically significant increase in the neutrophil count immediately after in comparison to before exercise in both groups. The lymphocyte counts significantly increased in both groups when values immediately after were compared to before exercise and for the comparison of values 1 h after versus immediately after exercise. The results suggest that the experimental protocol affected the immune system. These findings were corroborated by a previous study [22]. The elevation of neutrophils and lymphocytes into the bloodstream is probably mediated by the action of cortisol, catecholamines, and IL-6 [23,24].

The elevation in the number of circulating total leukocytes with the increase in neutrophils, monocytes, and lymphocytes including B and T cells is observed after intense physical activity. Therefore, it has been suggested that exercise represents a physical stress that is able to modify the counting of immune cells in peripheral blood circulation. It is not possible to assume the biological relevance of the increase in the number of circulating leukocytes. There were no significant differences observed between both groups.

Street running, of medium or high intensity, can generate mechanical stress, resulting in micro-injuries to muscle tissue [25]. This damage can be measured by the production of CK [26]. CK release triggers an inflammatory response that activates the transcription factor-kappa B [8]. Corroborating with findings in the literature, there was an increase in the concentration of CK in blood after performing the running protocol [27]. However, scientific evidence [4,28,29] suggests that curcumin might protect the muscle tissue, reducing the damage caused by running. We believed that this protective effect could influence the inflammatory response, i.e., less muscle damage, less inflammation. However, the muscle damage elicited by our experimental protocol was similar in both groups. This finding differs from the study of McFarlin et al. [30], in which curcumin reduced CK concentrations after a strength training protocol on the leg press apparatus.

In the present study, the experimental protocol promoted a significant increase in all cytokines evaluated in placebo-supplemented volunteers, indicating an inflammatory response induced by physical exercise. The increase in IL-2 concentration was corroborated by the results of a previous study (Konrad et al. 2011). The authors subjected 20 individuals to a treadmill running protocol at an intensity of 75% of maximal VO2 for 2 h. On the other hand, the supplementation of curcumin associated with piperine significantly reduced the concentration of IL-2 1 h after exercise in comparison to before exercise. Additionally, the comparison of both groups at 1 h after exercise showed lower levels of IL-2 in CPG than in PG. This result may be related to the ability of curcumin to inhibit the binding of IL-2 to its alpha receptor, reducing the production of the cytokine [31].

The levels of TNF-α and IFN were significantly increased in placebo-supplemented volunteers at 1 h after exercise. This finding was previously reported in studies with runners and may be related to exercise-induced muscle damage, promoting an inflammatory response [32,33]. In contrast, the supplementation of curcumin with piperine reduced TNF-α production, when comparing levels before and 1 h after exercise. Moreover, the comparison of both groups 1 h after exercise showed a significant reduction in TNF-α in the CPG. In regard to IFN concentrations, no changes were detected for values before and 1 h after exercise in the CPG. However, there was a significant difference between the groups at 1 h after exercise. A possible explanation for these findings is that curcumin can avoid the increase in TNF-α and IFN concentrations, probably via the inhibition of NF-kb [4,34].

The concentrations of IL-6 were significantly increased after 1 h after exercise in comparison to values before in the placebo-supplemented group. However, we did not observe any change in the CPG. In addition, IL-6 concentrations were significantly higher in PG than in CPG at 1 h after exercise. The increase in IL-6 concentration was already reported in a study with runners after an uphill run [35]. The elevation of circulating IL-6 post-exercise may be due to muscle resident macrophages activated at the site of tissue injury [36]. The increased levels of IL-6 are possibly associated with the elevation of TNF-α, which might stimulate IL-6 production [36,37]. However, we did not notice any change in IL-6 concentration in the CPG, suggesting a possible inhibitory action of curcumin supplementation. Curcumin can inhibit NF-kb, limiting IL-6 production [5,13]. Possibly, an IL-6 regulation pathway may be related to the ability of curcumin to inhibit NF-kb, which may reduce the production of pro-inflammatory cytokines, which may influence the production of IL-6 with a regulatory characteristic [36].

Although no significant difference was observed between the groups, the concentration of IL-10 significantly increased 1 h after exercise in comparison to values before in placebo-supplemented volunteers. This change was expected as a response to the increased concentration of inflammatory cytokines in this group. The inflammatory process produced by physical exercise induces the release of IL-10, an anti-inflammatory cytokine [38]. We hypothesize that the anti-inflammatory action of curcumin associated with piperine prevented the expression of inflammatory cytokines.

Intense physical exercise leads to a relevant inflammatory response mainly characterized by the mobilization of leukocytes and an increase in circulating inflammatory mediators produced by immune cells and directly from the active muscle tissue. Both positive and negative effects on immune function and susceptibility to minor illness have been observed following different types of training protocols. While engaging in moderate activity may enhance immune function above sedentary levels, excessive amounts of prolonged, high-intensity exercise may impair immune function. Thus, the aim of the present study was to study the effects of curcumin and piperine in response to a fatiguing exercise protocol. The findings showed that the supplementation was able to modulate the plasmatic cytokines levels which would be helpful in a high level of training and competition.

An important limitation of this study is the short time of supplementation, only 7 days, and we investigated only an acute effect of this supplementation. As a future perspective, we intend to investigate chronic supplementation of curcumin on performance parameters, inflammation markers, and the adaptive response to induced exercise.

## 5. Conclusions

Supplementation with 500 mg of curcumin associated with 20 mg of piperine for seven days was able to inhibit the inflammatory response after an experimental protocol of running on a treadmill until voluntary fatigue. However, the supplementation was not able to improve physical performance, protect against muscle damage, and alter the production of immune system cells.

## Figures and Tables

**Figure 1 biology-11-00573-f001:**
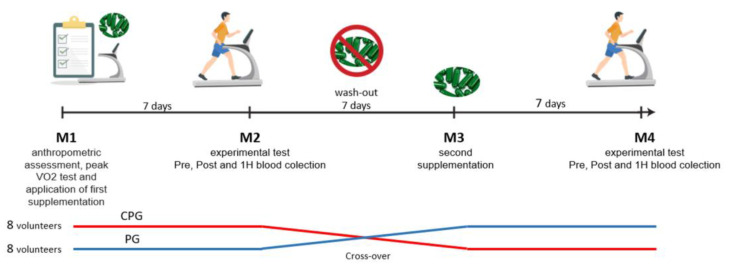
Experimental design.

**Figure 2 biology-11-00573-f002:**
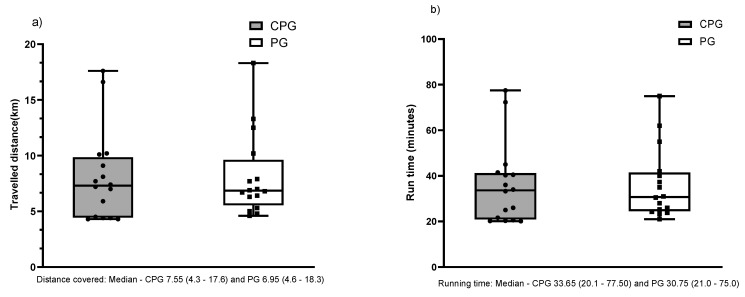
Graphs representing the comparison between the CPG and GP groups in the following variables: (**a**) distance and (**b**) running time. Values show in median and IQR.

**Figure 3 biology-11-00573-f003:**
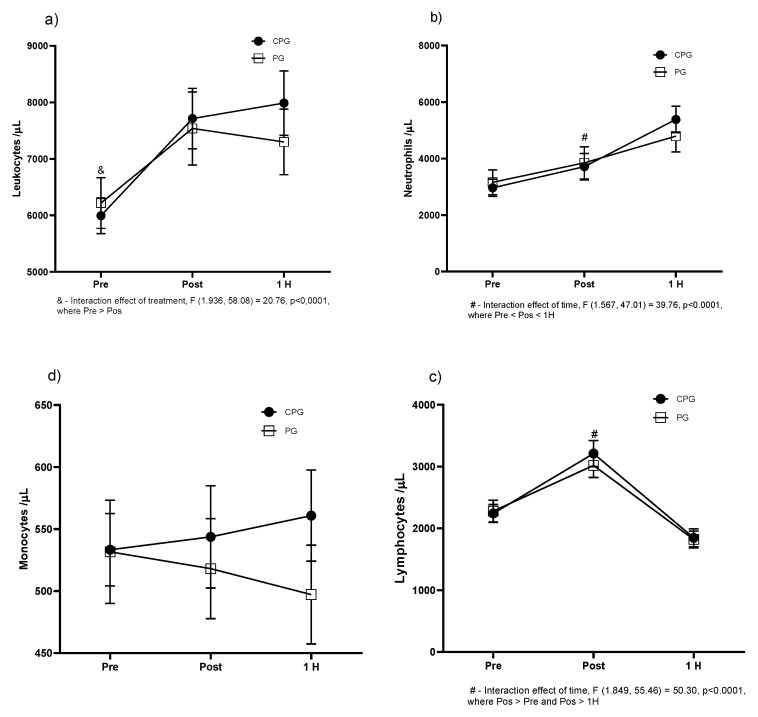
Graphs representing the leukocyte count across time: pre, post, and 1 h after performing the experimental running protocol. (**a**) Total leukocytes; (**b**) neutrophils; (**c**) monocytes; (**d**) lymphocytes. Values represented by the mean ± SD, (#) interaction (effect time), (&) interaction (time × treatment).

**Figure 4 biology-11-00573-f004:**
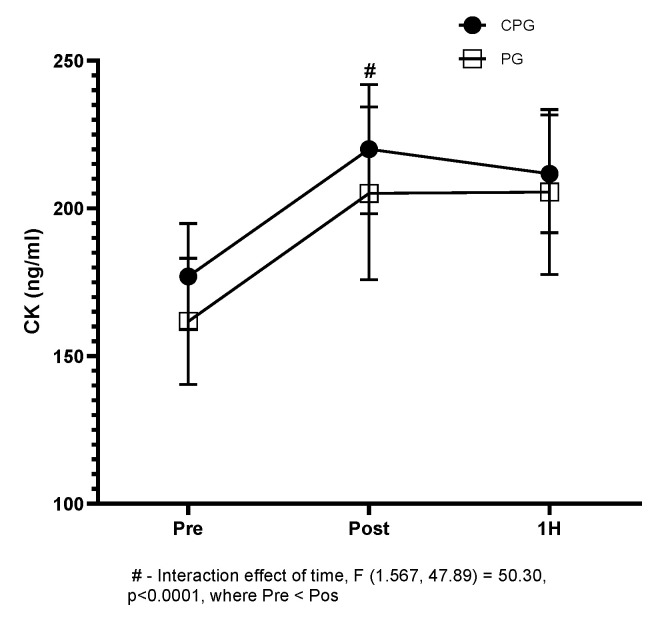
Chart represents the concentration (ng/mL) of creatine kinase (CK) following the experimental running protocol. Values represented by the mean ± SD, (#) interaction (effect of time).

**Figure 5 biology-11-00573-f005:**
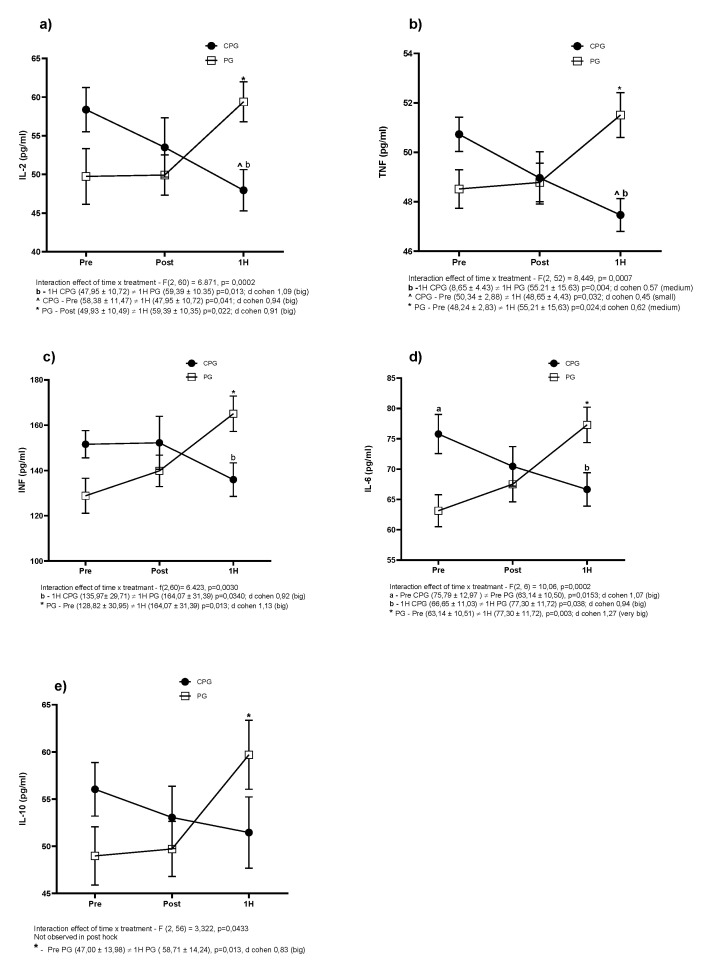
Circulating levels of interleukin-2 (IL-2) (**a**), tumor necrosis factor-alpha (TNF-α) (**b**), interferon (IFN) (**c**), interleukin-6 (IL-6) (**d**) and interleukin-10 (IL-10) (**e**). Values represented by the mean ± SD, *p*-value, and effect size.

## Data Availability

The data presented in this study are available on reasonable re-quest from the corresponding author.

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
