# Peer review of "The Curcumin Supplementation with Piperine Can Influence the Acute Elevation of Exercise-Induced Cytokines: Double-Blind Crossover Study"

_biology, 2022, doi:10.3390/biology11040573_

Round 1
Reviewer 1 Report
Comments on the individual sections are provided below.
Title
Curcumin Supplementation with Piperine Alters Inflammatory Response in Runners: Double-blind Crossover Study
In my opinion, the statement that "Curcumin Supplementation with Piperine Alters Inflammatory Response in Runners" is inadequate to the study results obtained.
Abstract:
line 38: mean age of 36.17 ± 9.05 - age is given in full years e.g. 36.0
Results:
curcumin and piperine 46 supplementation was able to reduce the concentrations of IL-2 (1H CPG 47,95 ±10,72 x 1HPG 59,40±10,35) pg/ml, p=0,013 and TNF (1H CPG 48,65±4,43x 1HPG 55,21±15,63) p=0,004; 48 it prevented the increases of IFN (1H CPG 135,97±29,7x1H PG 165,07±31,40) p=0,034 and 49 IL-6 (1H CPG 66,65±11,03x1H PG 77,30±11,71) p=0,038, showing the anti-inflammatory 50 effect of the supplementation;
The results of the study are presented in an unclear way and do not show any anti-inflammatory effect of curcumin supplementation with Piperine.
Conclusions:
Although curcumin supplementation had no effect on physical performance, immune cell counts or muscle damage, it was able to modulate exercise-induced inflammation.
The research results are not adequate for the obtained research results.
Introduction
Firstly, the Introduction does not sufficiently set out the rationale for the study.
If curcumin supplementation is to help athletes regenerate, parameters should be measured not only one hour after exercise, but 24 and 48 hours, even after 7 days after exercise.
line 62: In this regard, the association between curcumin and piperine may present an ergogenic aid in sports performance, basing on their antioxidant, antifungal, anti-inflammatory and antitumor functions [5,6]
What ergogenic support has to do with the anti-cancer effects of curcumin and piperine ?
line 75: often called tumor necrosis factor alpha or TNF-α
Materials and Methods
Study design and ethical approval
line 86: the average age is given in whole numbers 36 years
There is no information about the training experience of the runners, no history of infections and no injuries in the last 4 weeks prior to the study.
Study protocol
To check if the physical activity in both groups was similar during the intervention VO2max should be measured after the end of the study
2.5 Statistical analysis
General point; more detail on the test outcomes is needed. Exact p-values would be helpful or the effect of time, group and time x group if ANOVAs are considered.
Results
You must report the actual P values rather than simply writing NS.
The results in the figures are repeated in the tables
Table 1.
Incorrect description of table 1 . Incomprehensible descriptions of test results.
The effect size values are missing in figure 10.
- Discussion
line 337: The literature [4] suggests that associated a curcumin supplementation can improve physical performance in humans
The authors in the cited position do not claim that curcumin supplementation improves physical performance
- Conclusions
As exercise-induced inflammatory responses and oxidative stress are essential for muscle repair and regeneration and promote redox signalling pathways for adaptation, it is obvious that they must modulate the immune system. What is lacking in these studies is evidence that curcumin supplementation of runners alters the immune system response.
Conclusion should be revised
Author Response
Dear Editor,
Dear Reviewer
Prof. Dr. May Tang
Please find enclosed the revised version of our manuscript entitled “Curcumin Supplementation with Piperine Alters Inflammatory Response in Runners: Double-blind Crossover Study” to be considered for publication at Journal Biology.
We thank the Editorial Board for the opportunity to re-submit a revised version of our manuscript. We would also like to thank the reviewers for their helpful comments that significantly improved the manuscript. We addressed all comments as adequately as possible and highlighted the changes made in the newly submitted version in yellow. Please find attached a point-to-point response to the questions raised by the two reviewers.
We look forward to hearing from you in due course.
Yours Sincerely,

Reviewer 2 Report
The aim of this study was to evaluate the effect of supplementation of curcumin associated with piperine in runners after a treadmill running session until voluntary fatigue for the following parameters: physical performance, immune system cells, muscle damage, and circulating levels of inflammatory mediators.
Methodologically well-designed and very practical research, but the writing needs to be improved, especially in the discussion part.
Here are my contributions:
- The introduction should follow a common thread. The first paragraph, for example, talks about sport and the search for supplements with ergogenic effects and ends with a discussion of health-related effects, and then begins the second paragraph with a discussion of running. Paragraphs and ideas should be related and threaded together.
- In the protocol, wouldn't it make more sense to explain the protocol properly performed and then explain the materials and equipment used? order and coherence in the text.
- What was the purpose of performing a test at 80% of VO2max to fatigue, instead of doing an incremental test to fatigue again?
- The discussion is a bit descriptive in some sections, without elaborating on the reasons for the data obtained. For example, in the paragraph on line 244, there is no explanation or discussion of why there is an increase in neutrophils, or no differences between groups in the elevation of neutrophils and lymphocytes.
Author Response

(The authors gave the same response as above.)

Reviewer 3 Report
The authors examined the impact of one week of supplementation with curcumin combined with piperine, on physical performance, immune system cell counts, muscle damage and plasma levels of inflammatory markers after a treadmill running training session. The authors concluded that consuming the curcumin/piperine may reduce the exercise-enhanced blood inflammatory markers, whereas it did not change immune system cells, physical performance, and muscle damage after a running session until exhaustion.
Major comments
- This reviewer could not understand the novelty and rationale of this study. The authors state “In this regard, the association between curcumin and piperine may present an ergogenic aid in sports performance, basing on their antioxidant, antifungal, anti-inflammatory and antitumor functions (L62-65)”. What is the research question (i.e., problem)? What is the new finding(s) of this study? Why are those new findings important/advantages for runners (How important)?
- Why did the authors need to examine using a one-week intervention? In this context, why did the authors use 250 mg of curcumin with 10 mg of piperine extract? Please explain the rationale for choosing the intervention of this study.
- The participants were athletes because VO2peak was approx 60. Probably, the participants performed regular exercise training. The authors state “Subjects were instructed to consume 1 capsule at lunch and 1 capsule at dinner for seven consecutive days.”, but why? This reviewer guesses that the ingestion time against training is most important to manipulate exercise performance and muscle damages if curcumin/piperine acutely improves attenuates the inflammatory response after a single bout of exercise. As well as physical activity/exercise, the kinds of meals/nutrients may also be the important factors on physical performance and muscle damage. However, the authors did not check these variables (only 24 h prior to the first experimental trial), whereas you recommended to abstain from strenuous exercise in the 48 h before the exercise test and requested that physical training and food intake were kept consistent. If the authors did not have the data of activity/diet histories during the one-week intervention (e.g., PAQ/DHQ etc.), those are big limitations of this study.
- In statistics, the number of cells of the immune systems was analyzed using the Friedman test. Was that a non-normally distribution? Please mention it. If a normal data distribution was not confirmed, the data should be expressed as the median (IQR).
- In Figure 2, the authors explained the data as mean and SD. However, these box-whisker plots indicated median (IQR and max/min). Please correct it. In this context, the box-whisker plot should use for data of non-normally distribution and should be analyzed by the Friedman test.
- For all results using ANOVA, the authors must indicate the significant main effect/interaction before performing post-hoc tests. If there was no significant interaction, the authors can not do the specific post-hoc test, even though there was a significant main effect. Then, you can only perform the post-hoc test within the main effect. Please correct all.
- The authors state “The most important findings of this study were the administration of curcumin associated with piperine reduced the concentrations of IL-2, TNF and IFN, and maintained IL-6 and IL-10 at basal levels after the experimental protocol.”, but why/how important? This reviewer could not understand the importance of this finding. What is the contribution to science?
- For discussion, the authors JUST compared between conflict results of this study and the previous study. That’s why this reviewer would like to say “so what”. The different phenomena between this study and the previous study should be carefully explained using previous findings. Also, the authors should indicate e.g., “perspective” to clear the rationale and importance of this study.
- It seems like plasma levels of inflammatory markers at Pre in CPG were higher than in PG. How was P-values between CPG and PG at Pre? Why did one week of supplementation with curcumin combined with piperine increase the inflammatory markers at baseline (Is really that good results for sports performance and health)?
Minor
- In the abstract, the authors should mention the results of physical performance, immune cell counts, and muscle damage because you mentioned these variables in the conclusion.
- The authors state “At the second meeting, after the previous ingestion of supplementation, (L120)”. This time point is very important, so please indicate “XX min after”.
- Table 1, the units must be explained.
Author Response

(The authors gave the same response as above.)

Round 2
Reviewer 1 Report
After the correction, the article has become much easier to read. I realise that studies in which supplementation of individuals is used are always difficult and the response of different organisms is usually ambiguous. Therefore, conclusions should be drawn very carefully.
Author Response
Dear Reviewer
Please find enclosed the revised version of our manuscript entitled “The Curcumin Supplementation with Piperine can inhibit the acute elevation of cytokines exercise-induced: Double-blind Crossover Study” to be considered for publication at Journal Biology.
We thank the Editorial Board for the opportunity to re-submit a revised version during this round 2. We would also like to thank the reviewers for their helpful comments that significantly improved the manuscript. We addressed all comments as adequately as possible and highlighted the changes made in the newly submitted version in yellow. Please find attached a point-to-point response to the questions raised by the reviewers.
We look forward to hearing from you in due course.
Yours Sincerely,

Reviewer 2 Report
The authors have responded and improved the manuscript taking into account the contributions made. Congratulations.
Author Response

(The authors gave the same response as above.)

Reviewer 3 Report
Title; ‘elevation o cytokines’ should be ‘elevation of cytokines’.
This reviewer still has some concerns about statistics.
For Figure 2, the authors explain ‘normal data distribution’ because you performed the Student t-test (parametric test). However, you indicated the box-whisker plot with median (IQR). Why? Generally, if you need to show as median (IQR and max/min) instead of mean ± SD based on ‘non-normal data distribution’, you should perform the Wilcoxon test (non-parametric test). It seems like figure 2 may be non-normal data distribution according to the individual plots.
If the authors confirmed the main effect for time, please perform the post-hoc test against the main effect to detect the physiological meaning and please mention it.
‘Main effect of time × treatment’ should be ‘interaction (time × treatment)’.
Please discuss the impact of the one-week intervention on IL-6 (and INF) at rest because of the big effect size between CPG and PG at Pre. In addition, the amounts of the increases in cytokines are almost the same as compared with CPG at Pre. According to your answer, is this the negative impact? Probably, chronic rather than acute impact may be more important to understand physiological response for health and performance.
Probably, the results of 2-way ANOVA are complicated to interpret the physiological meaning. Please consider estimating the incremental/decremental AUC despite a few times points. However, if you will add the AUC, the authors must keep indicating the pre values.
The P-value of INF at pre was 0.0805. How was the power and effect size? If the statistical power is insufficient, this should be mentioned with the explanation of power calculation in the limitation section (Please add a new limitation section with some problems).
The authors state ‘The most important findings of this study were the administration of curcumin associated with piperine can inhibit the elevation of plasmatic level of some cytokynes’ but the results of IL-2 and TNF indicate ‘decrease’, not inhibit. Please reconsider it, including the title. It seems like homeostasis is broken, namely, exercise should increase/maintain circulating cytokines. In terms of PHILOSOPHY, why do the cytokines decrease in response to exercise?(Are the lack of cytokines in response to exercise (i.e., muscle damage) good for muscle repair etc.?) The authors have mentioned the possible mechanism, but please discuss it too.
Author Response

(The authors gave the same response as above.)
